# Expression Profiles of GILZ and Annexin A1 in Human Oral Candidiasis and Lichen Planus

**DOI:** 10.3390/cells11091470

**Published:** 2022-04-27

**Authors:** Mahmood S. Mozaffari, Rafik Abdelsayed

**Affiliations:** Department of Oral Biology and Diagnostic Sciences, The Dental College of Georgia, Augusta University, Augusta, GA 30912, USA; rabdelsa@augusta.edu

**Keywords:** human, GILZ, Annexin A1, immunohistochemistry, oral lichen planus, oral candidiasis

## Abstract

Adrenal glands are the major source of glucocorticoids, but recent studies indicate tissue-specific production of cortisol, including that in the oral mucosa. Both endogenous and exogenous glucocorticoids regulate the production of several proteins, including the glucocorticoid-induced leucine zipper (GILZ) and Annexin A1, which play important roles in the regulation of immune and inflammatory responses. Common inflammation-associated oral conditions include lichen planus and candidiasis, but the status of GILZ and Annexin A1 in these human conditions remains to be established. Accordingly, archived paraffin-embedded biopsy samples were subjected to immunohistochemistry to establish tissue localization and profile of GILZ and Annexin A1 coupled with the use of hematoxylin–eosin stain for histopathological assessment; for comparison, fibroma specimens served as controls. Histopathological examination confirmed the presence of spores and pseudohyphae for oral candidiasis (OC) specimens and marked inflammatory cell infiltrates for both OC and oral lichen planus (OLP) specimens compared to control specimens. All specimens displayed consistent and prominent nuclear staining for GILZ throughout the full thickness of the epithelium and, to varying extent, for inflammatory infiltrates and stromal cells. On the other hand, a heterogeneous pattern of nuclear, cytoplasmic, and cell membrane staining was observed for Annexin A1 for all specimens in the suprabasal layers of epithelium and, to varying extent, for inflammatory and stromal cells. Semi-quantitative analyses indicated generally similar fractional areas of staining for both GILZ and Annexin A1 among the groups, but normalized staining for GILZ, but not Annexin A1, was reduced for OC and OLP compared to the control specimens. Thus, while the cellular expression pattern of GILZ and Annexin A1 does not differentiate among these conditions, differential cellular profiles for GILZ vs. Annexin A1 are suggestive of their distinct physiological functions in the oral mucosa.

## 1. Introduction

Glucocorticoids exert well-known immunosuppressive and anti-inflammatory effects in a variety of conditions associated with dysregulation of immune and inflammatory responses [1,2]. Both exogenous and endogenous glucocorticoids mediate their multifaceted anti-inflammatory effects primarily through regulation of gene transcription and subsequent generation of a number of effector molecules. Prominent among them are the glucocorticoid-induced leucine zipper (GILZ) and Annexin A1 proteins [2,3,4,5,6,7]. GILZ mediates a myriad of effects of glucocorticoids, including inhibition of dendritic cell activity, inhibition of macrophage function, regulation of thymic selection, and T-helper (Th)-1 and Th-2 differentiation, increased regulatory T cell activity, and inhibition of inflammatory cytokine production, among others [1,2,3,4]. Similarly, Annexin A1 regulates the impact of glucocorticoids on leukocyte trafficking, among others, thereby contributing to their anti-inflammatory effects [5,6,7]. These effects of GILZ and Annexin A1 are of clear relevance and importance for oral conditions for which dysregulation of immune and inflammatory responses plays a pathogenic role and/or is associated with such disorders.

Oral lichen planus (OLP), considered a potentially malignant disorder, is a subtype of this condition that affects oral mucosa with a prevalence of about 0.5–2% [8,9]. Clinical manifestations of OLP include reticular, papular, plaque, erythematous or atrophic, erosive, and bullous presentations, while in some cases, they may manifest as a combination of such presentations. Importantly, malignant transformation (1.14%) has been reported for OLP, but it is also suggested that the potential for malignant transformation is likely underestimated, in part, due to the application of restrictive diagnostic criteria (e.g., epithelial dysplasia) [10]. Current treatment options for OLP include topical glucocorticoids and other immunomodulatory agents (e.g., tacrolimus) [11]. Interestingly, a recent study reported that injections of platelet-rich fibrin into OLP lesions of human subjects exert similar effects (i.e., reduction of lesion size and visual analog scale score) to those of triamcinolone acetonide injections [12].

Oral candidiasis (OC), on the other hand, is an opportunistic infection commonly caused by the overgrowth of *Candida albicans* [13]. It is estimated that about 30–60% of healthy adults carry this microorganism within their oral cavity, which exists as commensal colonization. However, pathological colonization of *Candida* species is related to several factors including extremes of age, metabolic disorders (e.g., diabetes mellitus), compromised host defenses, medications (e.g., antibiotics, corticosteroids), radiotherapy, salivary gland hypofunction, and malnourishment [11,13]. An acute pseudomembranous form of OC is a common presentation, and other manifestations include chronic erythematous candidiasis, acute or chronic atrophic candidiasis, and chronic hyperplastic candidiasis. Importantly, cases of chronic OC can clinically mimic leukoplakia or dysplasia. Thus, while antifungal agents remain the mainstay of therapy for OC [11], histopathological assessment of the lesion is required for microscopic diagnosis and to guide patient management.

The focus of this cross-sectional study was to establish expression profiles of GILZ and Annexin A1 in OLP and OC as prevalent oral inflammatory lesions, given that, aside from the adrenal glands as a primary source of endogenous glucocorticoids, other tissues such as the oral mucosa also produce cortisol [4,14,15]. Thus, we tested the hypothesis that the staining pattern and intensity for GILZ and Annexin A1 are similar for OLP and OC. Accordingly, GILZ and Annexin A1 immunoprofiles and semi-quantitative assessment of staining in these lesions were contrasted with those of oral fibromas as the control condition.

## 2. Materials and Methods

This study used archived paraffin-embedded biopsy samples of patients who either presented to community dental professionals or were evaluated in the Department of Oral Biology and Diagnostic Sciences, Section of Oral Maxillofacial Pathology, of the Dental College of Georgia. All patients’ personal identifiers were removed prior to the use of archived samples for this study that was deemed exempt from review by the Institutional Review Board. These patients were initially evaluated for clinical presentations of their oral lesions, followed by subsequent histopathological assessment of their biopsy specimens, which revealed OLP (*n* = 10) and OC (*n* = 10); tissues specimens displaying fibroma (*n* = 4) and papilloma (*n* = 1) were used as control. Table 1 summarizes demographic information, anatomical site of the lesion, and clinical impression/diagnosis of the oral presentation (i.e., prior to histopathological evaluation).

For histopathological examination, paraffin-embedded tissue specimens were cut in 5 μm thickness, mounted on glass slides, and de-paraffinized in a Leica Auto-Stainer XL; a Citric Acid based Antigen Unmasking Solution (Vector Laboratories, Burlingame, CA, USA) was used for antigen retrieval [4]. Thereafter, tissue sections were treated with 0.3% hydrogen peroxide for 30 min at room temperature, washed in water, followed by incubation in Blocking solution (2.5% horse serum, 1% bovine serum albumin, 0.5% Triton X-100) for at least 1 h at room temperature. Each primary antibody was diluted (1:100) in Blocking solution and incubated with tissue sections overnight at room temperature; GILZ mouse monoclonal antibody was obtained from LifeSpan Biosciences, Inc. (LS-B4313), while rabbit monoclonal antibody [EPR19,342] to Annexin A1/ANXA1 (ab214,486) was purchased from Abcam. Tissue sections were then washed twice in phosphate-buffered saline and incubated for 1 h at room temperature with horseradish peroxidase-conjugated secondary antibody (Vector Laboratories, Burlingame, CA, USA), followed by 3,3′-diaminobenzidine (DAB) staining using the ImmPACT DAB Substrate Kit (Vector Laboratories, Burlingame, CA, USA). Then, slides were counter-stained with hematoxylin and mounted with a mounting medium. As positive control for antibodies, human mammary tissue was used to establish staining for GILZ, while human tonsil was used for Annexin A1 staining. For histopathological assessment, tissue specimens were subjected to hematoxylin–eosin (H&E) staining, while periodic acid–Schiff (PAS) stain was used to confirm fungal organisms.

The Image J Fiji software was utilized for semi-quantitative assessment of immunohistochemical staining based on a previously described protocol [16]. The protocol involves deconvolution of immunohistochemistry images followed by assessment of DAB staining, using mean grey intensity, and normalization to the nucleus. We also measured the fractional area of staining.

### Statistics

Semi-quantitative data are reported as means ± SEM for each condition. All data were analyzed using the analysis of variance followed by Duncan’s post hoc test to establish significance (*p* < 0.05) among experimental conditions.

## 3. Results

Table 1 summarizes relevant features of subjects whose biopsy specimens were used in this study. Figure 1 shows H&E staining for three tissue specimens for each condition (scale bar: 100 μm), and Figure 2 shows PAS staining for one OC specimen to verify the presence of fungal infection (scale bar: 50 μm). Figure 3 and Figure 4 show immunohistochemistry images of proteins of interest; each figure shows images of three specimens for each condition (scale bar: 100 μm). For greater ease of identification of immunohistochemical features, Figure 5 shows higher magnification images for one biopsy specimen in each category (scale bar: 50 μm). Figure 6 shows semi-quantitative data for the fractional area of staining (A) and normalized staining (B) for experimental conditions.

### 3.1. Histopathological Assessment

#### 3.1.1. Control Specimens

A total of five cases (four fibromas and one papilloma as clinical impression) were used as control. The surface epithelium of these cases is essentially intact with minimal keratinization. The nodular fibrous lesions lacked subepithelial inflammation (Figure 1).

#### 3.1.2. Oral Candidiasis (OC)

Samples show mucosal sections with parakeratinized surface epithelium with supporting connective tissue (Figure 1 and Figure 2). The typical features in this group included shaggy surface parakeratin that supported neutrophilic aggregates intermixed with spores and pseudohyphae of *Candida albicans.* The basal cell layer was intact and exhibited basilar hyperplasia in some of the cases. The fungal organisms could be identified using PAS stain; the image for one case is shown in Figure 2. The subepithelial connective tissue supported an intense, diffuse and mixed infiltrates of lymphocytes, plasma cells, histiocytes, and occasionally neutrophils.

#### 3.1.3. Oral Lichen Planus (OLP)

H&E-stained sections show oral mucosa with parakeratinized or orthokeratinized surface epithelium and supporting connective tissue (Figure 1). The surface epithelium varied in thickness in some samples while atrophied in others. Characteristic features included basal cell layer vacuolar degeneration with eosinophilic apoptotic bodies, known as “*Civatte bodies*”, identified at the epithelium–connective tissue interface. However, there was no evidence of dysplasia or cellular atypia in any of the cases. A subepithelial band of lymphohistiocytic infiltrate is seen, which blended with the base of the surface epithelium.

### 3.2. GILZ Immunohistochemistry

#### 3.2.1. Control Specimens

All sections of fibromas showed positive nuclear staining throughout the full thickness of the epithelium (Figure 3 and Figure 5). Additionally, stromal cells, including fibroblasts and vascular endothelium, exhibited positive nuclear reactions.

#### 3.2.2. Oral Candidiasis

Sections of oral mucosa with candidiasis showed a diffuse positive nuclear reaction in the epithelial cells throughout the surface epithelium (Figure 3 and Figure 5). However, the candida organisms were not stained with the GILZ antibody. The subepithelial infiltrate showed mixed populations of predominantly small size cells with a nuclear positive reaction, consistent with lymphocytes, and unstained larger size cells, histiocytes, and plasma cells. Staining of stromal cells, including fibroblasts and vascular endothelium, exhibiting nuclear reaction was also observed.

#### 3.2.3. Oral Lichen Planus

Sections stained with antibody for GILZ consistently showed a diffuse positive nuclear reaction in the epithelial cells throughout the surface epithelium (Figure 3 and Figure 5). No distinct cytoplasmic staining was observed. The subepithelial infiltrate showed mixed populations of predominantly small size cells with a nuclear positive reaction, consistent with lymphocytes, and unstained larger size cells, the histiocytes. Staining of stromal cells, including fibroblasts and vascular endothelium, exhibiting nuclear reaction was also observed.

### 3.3. Annexin A1 Immunohistochemistry

#### 3.3.1. Control Specimens

All sections of fibromas showed positive Annexin A1 nuclear staining in the epithelium, while additional cytoplasmic and cellular membrane staining was observed in some of the epithelial cells in a patchy distribution (Figure 4 and Figure 5); however, for most sections, the basal layer(s) showed minimal or no stain. Stromal cells, including fibroblasts and vascular endothelium, exhibited positive nuclear reaction.

#### 3.3.2. Oral Candidiasis

Sections stained with Annexin A1 antibody showed diffuse and prominent nuclear reactivity of the suprabasal epithelium, but the basal layers were spared from staining (Figure 4 and Figure 5). Additionally, cytoplasmic and cell membrane staining was observed in some of the suprabasal epithelial cells in a patchy distribution. The subepithelial inflammatory cell infiltrate showed mixed populations of predominantly small size cells with a nuclear positive reaction, consistent with lymphocytes, and larger size unstained cells, most likely histiocytes. Staining of stromal cells, including fibroblasts and vascular endothelium, exhibiting nuclear reaction was also observed.

#### 3.3.3. Oral Lichen Planus

Sections stained with Annexin A1 antibody showed diffuse nuclear reactivity throughout the surface epithelium, whereas additional cytoplasmic and cellular membrane staining were observed in some of the epithelial cells in a patchy distribution (Figure 4 and Figure 5). However, sparring of basal and suprabasal layers could not be clearly discerned. The subepithelial inflammatory cell infiltrate showed mixed populations of predominantly small size cells with positive nuclear reaction, consistent with lymphocytes, and larger size unstained cells, most likely histiocytes. Staining of stromal cells, including fibroblasts and vascular endothelium, exhibiting nuclear reaction was also observed.

#### 3.3.4. Semi-Quantitative Analysis

Figure 6 shows the results of the semi-quantitative assessment of staining for experimental groups. As shown in panel A, fractional areas for GILZ and Annexin A1 staining were generally similar among the groups, albeit the differential for GILZ fractional area of staining was marginally significant between the control and OLP groups (*p* = 0.061). On the other hand, panel B shows DAB staining for proteins of interest, normalized to a number of nuclei, as has been described in detail earlier [16]. The results indicate a significant reduction for GILZ normalized staining for OC and OLP compared to the control group; however, Annexin A1 normalized staining was generally similar among the groups.

## 4. Discussion

This study shows an abundant expression of GILZ and Annexin A1 in oral biopsy specimens of humans, with heterogeneous demographics and histopathological diagnosis of OC, OLP, or fibroma. Nonetheless, while epithelial GILZ immunostaining is confined to the nuclei, Annexin A1 immunostaining is evident for nuclei, cytoplasm, and plasma membrane. Importantly, the lack of Annexin A1 immunostaining is a feature of basal layers of epithelium, albeit less discernable for OLP but more marked for OC specimens. Further, the inflammatory cells of OC and OLP specimens and stromal cells of all specimens showed varied expression of GILZ and Annexin A1, primarily in nuclei. While staining areas for GILZ and Annexin A1 were generally similar among the groups, normalized staining for GILZ, but not Annexin A1, was reduced for OC and OLP groups compared to the control group. To our knowledge, this is the first demonstration of differential expression profiles for GILZ and Annexin A1 in inflammatory oral lesions, thereby suggesting distinct functional roles in the human oral mucosa.

The role of GILZ in several disorders associated with mucosal inflammation has been the subject of investigation. Persistent inflammation of sinonasal mucosa is a characteristic feature of chronic rhinosinusitis without or with nasal polyps. For both conditions, suppression of GILZ mRNA and protein expressions has been reported compared to the upper airway mucosa of control patients. Further, patients with both conditions who were refractory to surgery displayed a greater decrease in GILZ expression in upper airway mucosa [17]. Chronic rhinosinusitis with nasal polyps is frequently associated with asthma, and accompanying bacterial infection can aggravate the disease. These considerations led to the investigation of the impact of pre-incubation of nasal fibroblasts with lipopolysaccharide (LPS), which, among other effects, reduced GILZ expression, an effect likely explaining how bacterial infection of upper airways may limit the efficacy of glucocorticoid treatment [18]. In this context, activation of the prostacyclin receptor was shown to augment the ability of glucocorticoids to induce anti-inflammatory genes, including GILZ, in human airway cells. This observation may be of relevance for the treatment of airway inflammatory diseases with suboptimal response to or refractory to glucocorticoid treatment [19]. Of relevance to the respiratory system is the association of alcohol abuse with immunosuppression and infectious sequelae such as pulmonary infections. Interestingly, alcohol dose-dependently increased GILZ gene and protein expressions in primary human airway epithelial cells leading to the suggestion that GILZ may play a role in the anti-inflammatory and immunosuppressive effects of alcohol [20]. To gain insight into the role of GILZ in the gastrointestinal tract, an intestinal epithelial cell line (i.e., MODE-K cells) was infected with *T. gondii*. Treatment of infected MODE-K cells with corticosterone increased the level of 17 kDa GILZ isoform. Further, corticosterone-treated cells had decreased expression of several chemokines, while their expression was increased by siRNA-induced inhibition of endogenous GILZ production. Authors concluded that GILZ up-regulation during infection might serve as a mechanism to decrease epithelial cell responses and facilitate parasite replication [21]. However, utilizing the dextran sulfate sodium (DSS)-induced mouse model of colitis, treatment with TAT-GILZ, a cell-permeable GILZ fusion protein, after the onset of colitis improved gut permeability and ameliorated gut dysbiosis. These observations are suggestive of the promotion of an optimal environment for colonization of the mucosa surface by beneficial bacteria conducive to healing [22]. Others have shown increased GILZ expression in association with reduced ulceration and inflammation in the dexamethasone-treated hamster model of oral mucositis induced by 5-fluorouracil and trauma [23]. We have observed nuclear staining for GILZ throughout the full thickness of epithelium in tissue specimens of subjects, with heterogeneous demographics and very different etiopathogenesis with one of the conditions, i.e., OC, being of infectious etiology and both OC and OLP being associated with marked infiltration of inflammatory cells that, to varying extent, stain for GILZ as do stromal cells. While the fractional area of staining was marginally significant for the OLP compared to control specimens, normalized staining was significantly reduced for both OC and OLP specimens. One can speculate that the latter observation relates, in part, to the larger number of cells/nuclei in OC and OLP specimens, given the marked infiltration of inflammatory cells in those tissues. Nonetheless, the outcome is reduced staining, and thus protein abundance, in the microenvironment of the lesions. Given the well-documented anti-inflammatory effects of GILZ, we conjecture that reduction of GILZ in OC may be helpful (i.e., favoring pro-inflammatory activity) and compensatory for the presence of *Candida albicans* (i.e., to reduce fungal colonization and activity) while a similar reduction of GILZ in the microenvironment in OLP could exacerbate the underlying pathology. Thus, the therapeutic utility of topical glucocorticoids in OLP may relate to the upregulation of GILZ generation within the lesion. In this context, the advent of the cell-permeable GILZ fusion protein (i.e., TAT-GILZ) should help determine the role and therapeutic value of GILZ in OLP and OC [22].

With respect to Annexin A1, its expression has been investigated in normal and chronically inflamed nasal mucosa and nasal polyps of human subjects [24]. Accordingly, high expression of Annexin A1 was observed on the apical surface and cytoplasm of ciliated cells without staining of undifferentiated basal epithelial cells and goblet cells. Further, ductal epithelial cells of the glands of lamina propria showed intense cytoplasmic and nuclear stains, but acinar cells did not stain. Annexin A1 staining was also reported for infiltrating macrophages and polymorphonuclear cells. Interestingly, however, the pattern or the expression level of Annexin A1 was not affected in the epithelial cells and glands of normal and chronically inflamed (perennial rhinitis or polyps) nasal mucosa. Authors concluded that Annexin A1 expression in respiratory epithelium relates to the type of cells and their differentiation status rather than their inflammatory status [24]. The same group also explored expression patterns of Annexin A1 and Annexin A2, two structurally and phylogenetically related proteins, in tissue specimens from respiratory (nasal and laryngeal) and digestive (oral and pharyngeal) mucosa of non-cancer patients. While Annexin A1 was expressed in the more differentiated cells, Annexin A2 was expressed primarily in less differentiated cells, thereby suggestive of their different physiological functions in the aerodigestive tract [25]. Of therapeutic interest is a report that the Annexin A1 peptide mimetic, Ac2-26, exerted multifaceted inhibitory effects on airway inflammation and hyper-responsiveness in a rat model of asthma [26]. With respect to the gastrointestinal tract, chronic gastritis displays a high expression of Annexin A1 mRNA that likely contributes to the healing of gastric mucosal damage [27,28]. In inflammatory bowel disease (IBD), loss of Annexin A1 expression likely promotes inflammatory status, while an enhanced level of Annexin A1 may be predictive of the effectiveness of the therapeutic intervention [29]. Further, treatment with an Annexin A1-based tripeptide (MC-12) exerts a strong beneficial effect in animal models of human IBD [30]. In ulcerative colitis, Annexin A1 upregulation is a feature of subjects in remission, thereby suggestive of a significant role in promoting mucosal homeostasis [31]. Further, Annexin A1 secretion, by infiltrating neutrophils and macrophages, is a feature of a chemically-induced rat model of colitis [32]. On the other hand, Annexin A1-deficient animals have increased susceptibility to DSS-induced colitis in association with greater morbidity and mucosal injury that is ameliorated by an agonist for Annexin A1 receptors, thereby suggestive of protective and reparative properties of endogenous Annexin A1 on the intestinal mucosal epithelium [33]. Indeed, the interaction of Annexin A1 with its epithelial receptors linked to NADPH oxidase is believed to promote mucosal wound repair [34]. In this context, deficiency of Annexin A1 reportedly worsens the phenotype of cystic fibrosis, a condition characterized by abnormal fluid transport across secretory epithelia and chronic inflammation involving the lung, pancreas, and intestine [35]. Consistent with findings in human nasal mucosa [24], we observed nuclear, cytoplasmic, and cell membrane staining for Annexin A1 in the epithelium of all tissue specimens, a pattern different than that for GILZ staining, which was confined to nuclei; however, neither the fractional area of staining nor normalized staining for Annexin A1 clearly differentiated among the OC, OLP and fibroma specimens. Others have shown that in human normal oral mucosa, Annexin A1 staining is predominantly localized to the cell membrane; however, in oral epithelial dysplasia and oral squamous cell carcinoma specimens, cell membrane staining decreased, while nuclear staining increased, thereby indicating a role in malignant transformation [36]. Further, loss of Annexin A1 is a frequent and early event during head and neck carcinogenesis [37,38,39], while increased mucosal Annexin A1 expression is reported for gastric adenocarcinoma [27]. Collectively, these observations suggest that the role of Annexin A1 in maintaining epithelial differentiation is cell- and context-specific. In this context, we observed that (the undifferentiated) basal/suprabasal layer(s) of epithelium did not stain for Annexin A1, a finding more prominent for OC than control or OLP specimens; the finding in relation to OLP is likely related to liquefaction degeneration of basal and suprabasal layers of the epithelium which is a characteristic feature of this condition. Our observation is consistent with the role of Annexin A1 in cell proliferation and differentiation [40]. For example, Annexin A1 is implicated in hematopoietic stem cell/progenitor cell differentiation and favoring myeloid/granulocytic lineage, modulating T cell proliferation and differentiation as well as myoblast cell differentiation into skeletal muscle cells [41,42,43]. Importantly, robust expression of Annexin A1 in OC specimens is consistent with a report suggesting Annexin A1 as a strong candidate as an epithelial cell anti-candida effector protein, thereby contributing to keeping this organism in a commensal state. Nonetheless, this compensatory mechanism can be overwhelmed by other factors (e.g., low expression of E-cadherin, reduced CD8+ T cell infiltration), thereby increasing susceptibility to OC [44,45]. Importantly, in addition to its inhibition of phospholipase A2, Annexin A1 can be externalized/secreted, which, in turn, acts via autocrine and paracrine mechanisms to exert its physiological effects, including anti-inflammatory actions [5]. Thus, from a therapeutic perspective, the availability of Annexin A1-mimetics (e.g., Ac2-26) and Annexin A1 receptor agonists (e.g., Compound 17b and Compound 43) should facilitate the investigation of their efficacy (e.g., via topical application to oral mucosa) in OC and OLP [5].

In conclusion, despite heterogeneous demographics, staining patterns for GILZ and Annexin A1 in the oral mucosal epithelium of human subjects did not differentiate the inflammatory conditions of OLP and OC from the fibroma. Rather, distinct patterns of staining for these proteins are suggestive of their differential functional roles in the oral mucosa. Further, we observed a significant reduction in normalized staining for GILZ, but not Annexin A1, for OLP and OC compared to fibroma. Given the marked and multifaceted anti-inflammatory effects of GILZ, its reduction in the microenvironment of OC and OLP specimens could curtail the growth of *Candida albicans* in OC, while a similar reduction in OLP could exacerbate the inflammation associated with OLP, aspects that require further investigation. On the other hand, semi-quantitative analyses did not clearly differentiate among the groups in relation to Annexin A1; nonetheless, the availability of Annexin A-mimetics and receptor agonists should facilitate a better understanding of its role in OC and OLP.

## Figures and Tables

**Figure 1 cells-11-01470-f001:**
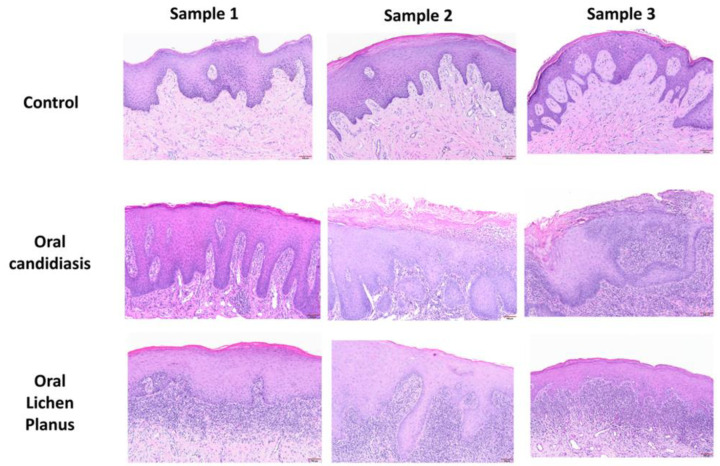
Panels show H&E staining of three tissue specimens from each experimental group confirming presence of marked immune and inflammatory cells in oral candidiasis and oral lichen planus compared to control tissues. Scale bar: 100 μm.

**Figure 2 cells-11-01470-f002:**
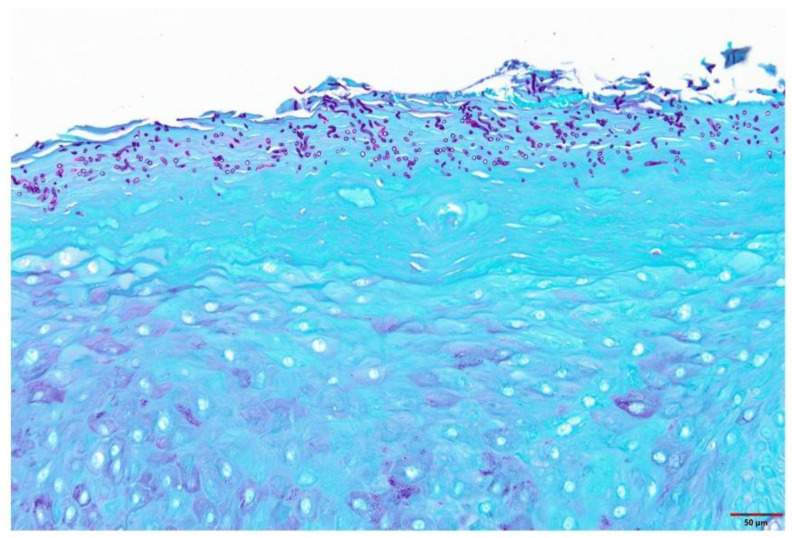
Panel shows image from one tissue specimen stained with PAS to show presence of *Candida albicans*. Scale bar: 50 μm.

**Figure 3 cells-11-01470-f003:**
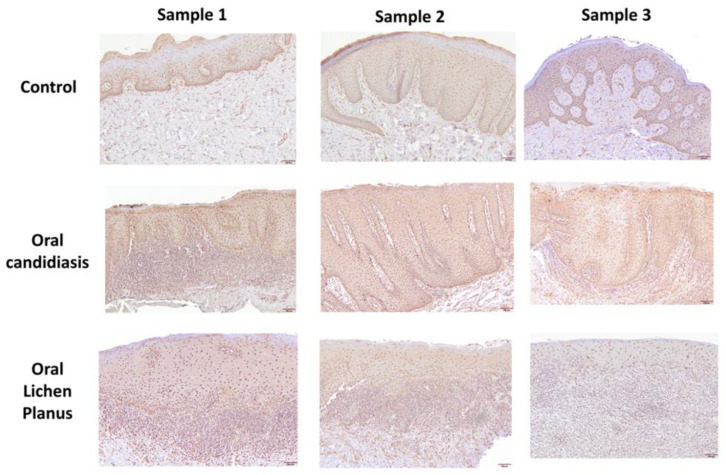
Panels show immunohistochemical staining for GILZ in three tissue specimens from each group of control, oral candidiasis, and oral lichen planus. For each condition, GILZ immunostaining was confined to nuclei. Scale bar: 100 μm.

**Figure 4 cells-11-01470-f004:**
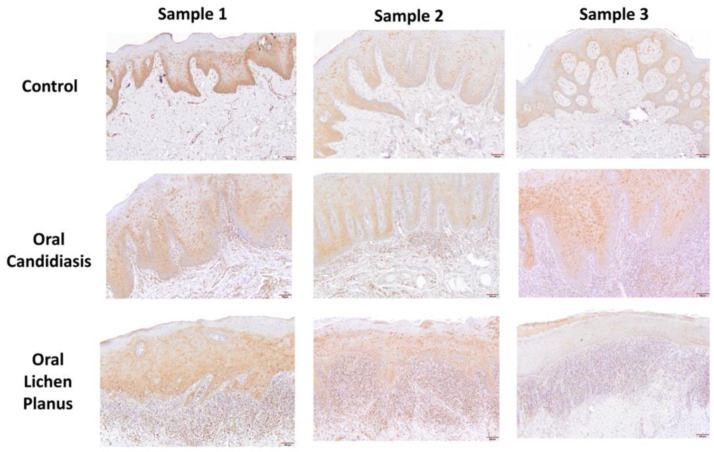
Panels show immunohistochemical staining for Annexin A1 in three tissue specimens from each group of control, oral candidiasis, and oral lichen planus. Each condition displayed nuclear, cytoplasmic and cell membrane staining for Annexin A1, albeit to varying extent as described under Results. Scale bar: 100 μm.

**Figure 5 cells-11-01470-f005:**
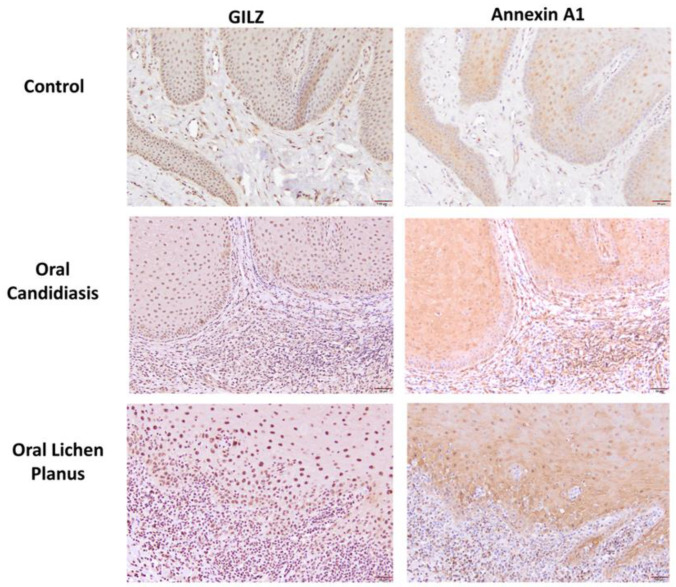
Panels show GILZ and Annexin A1 staining for one tissue sample from control, oral candidiasis, and oral lichen planus groups at higher magnification to better illustrate staining patterns. While GILZ immunostaining is confined to nuclei, Annexin A1 immunostaining is seen for nuclei, cytoplasm, and cell membrane. Further, sparring of Annexin A1 immunostaining is seen for basal layers of epithelium, although less discernable for OLP but more prominent for OC, compared to control, specimens. Scale bar: 50 μm.

**Figure 6 cells-11-01470-f006:**
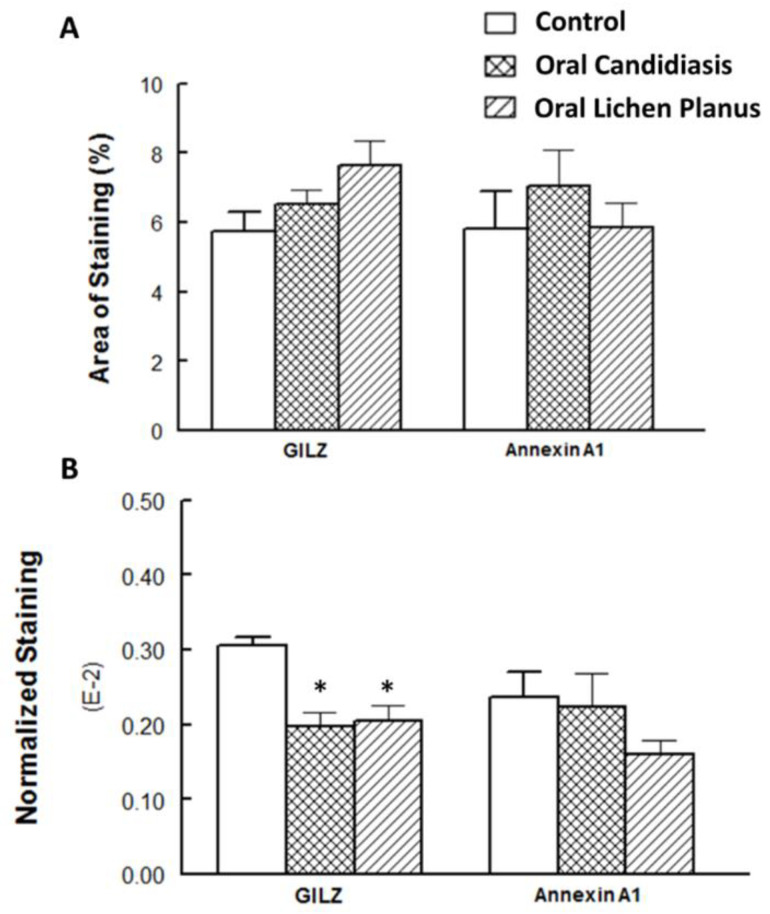
Panel (**A**) shows fractional area of staining, while panel (**B**) shows staining normalized to nuclei for proteins of interest. Data are means ± SEM for each condition; *n* = 5–10 specimens for each condition as indicated under Methods. * *p* < 0.05 compared to the control group.

**Table 1 cells-11-01470-t001:** Features of experimental subjects whose biopsy samples were used for this study. Accordingly, age, sex, ethnicity (if known), and anatomical site of lesions are included, as well as clinical impression/diagnosis prior to biopsy for histopathological assessment.

	Age (Years)	Sex	Ethnicity	Anatomical Site	Clinical Impression/Diagnosis
Control					
Patient 1	33	Male	Unknown	Maxillary ridge	Fibroma
Patient 2	21	Male	Unknown	Lower lip	Fibroma
Patient 3	76	Male	Unknown	Lower lip	Fibroma; mucocele
Patient 4	69	Female	Caucasian	Tongue	Fibroma
Patient 5	54	Male	Caucasian	Buccal mucosa	Papilloma
Oral Candidiasis					
Patient 1	56	Male	African-American	Buccal mucosa	Candidiasis; leukoplakia
Patient 2	37	Male	Caucasian	Buccal mucosa	Leukoplakia; lichen planus
Patient 3	36	Male	African-American	Tongue	Candidiasis
Patient 4	79	Female	Caucasian	Tongue	Hyperkeratosis; dysplasia; squamous cell carcinoma
Patient 5	66	Male	African-American	Buccal mucosa	Leukoplakia
Patient 6	82	Male	Caucasian	Buccal mucosa	Papilloma
Patient 7	61	Female	Unknown	Retromolar pad	Candidiasis; dysplasia
Patient 8	61	Female	Unknown	Palate	Candidiasis
Patient 9	44	Female	African-American	Tongue	Lichenoid mucositis; lichen planus
Patient 10	74	Male	Caucasian	Tongue	Ulcer; dysplasia
Oral Lichen Planus					
Patient 1	59	Female	Caucasian	Buccal mucosa	Lichen planus
Patient 2	69	Male	Caucasian	Buccal mucosa	Lichen planus
Patient 3	32	Male	Unknown	Tongue	Keratosis
Patient 4	32	Male	Caucasian	Buccal mucosa	Traumatic keratosis
Patient 5	55	Female	Unknown	Buccal mucosa	Lichen planus
Patient 6	67	Female	Caucasian	Hard/soft palate	Lichen planus
Patient 7	39	Male	Caucasian	Maxillary gingiva	Lichen planus
Patient 8	39	Female	Caucasian	Buccal mucosa	Lichen planus
Patient 9	58	Female	Caucasian	Buccal mucosa	Lichen planus
Patient 10	71	Female	Caucasian	Maxillary gingiva	Lichen planus

## Data Availability

The data presented in this study are available on request from the corresponding author.

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
