# Peer review of "Expression Profiles of GILZ and Annexin A1 in Human Oral Candidiasis and Lichen Planus"

_cells, 2022, doi:10.3390/cells11091470_

Round 1

Reviewer 1 Report

Manuscript ID: cells-1646169

Title: Expression Profiles of GILZ and Annexin-A1 in Human Oral Candidiasis and Lichen Planus

1.What is the main question addressed by the research?

To assess expression profiles of GILZ and Annexin A1 in OLP and OC as prevalent oral inflammatory lesions in oral lichen planus (OLP) and oral candidiasis (OC) using oral fibromas as control.

2.Is it relevant and interesting?

The article is relevant and interesting.

3.How original is the topic?

The topic is current.

4.What does it add to the subject area compared with other published material?

The authors have collected and analyzed a great deal of data.

5.Is the paper well written?

Yes, the article is well written.

6.Is the text clear and easy to read?

Yes, but minor English editing is required.

7.Are the conclusions consistent with the evidence and arguments presented?

Yes, the conclusions consistent with the evidence and arguments presented but further studies are necessary to confirm authors’ hypothesis.

8.Do they address the main question posed?

Yes, the Authors addressed the main question posed.

Other comments:

On my opinion the article is interesting and well written. However, I highlighted some minor issues:

  • English language: Minor spell check required.
  • Summary of abbreviations required.
  • Introduction: The Authors may improve this section on the theme of therapies for the treatment of OLP. Allow me to suggest a relevant references to include on the use of platelet-rich fibrin on the treatment of OLP: “https://doi.org/10.1007/s00784-020-03702-w”.
  • Materials and methods: This section has been properly prepared.
  • Results: This section is not clearly written. Please improve.
  • Discussion: This section has been properly prepared.
  • Conclusion: This section has been properly prepared.

After making the indicated changes, the article may be suitable for publication.

Thanks for the opportunity to review this manuscript.

Author Response

We are truly grateful to the reviewer for her/his very valuable comments and suggestions that we have carefully considered in revision of our manuscript; revisions are highlighted in BLUE for ease of identification.  We are delighted that the reviewer has found our submission to be timely and valuable and are thankful for his/her supportive comments.  Items requiring a response appear below.  

As an additional comment, we wish to indicate that we have revised Figure 2 to show the 50 μm (i.e., higher magnification) image, only.  [In other words, we have removed the 100 μm, lower magnification, image which appeared in our initial submission.]  We believe that the 50 μm image is sufficient to show candida infection.  We hope this revision is acceptable to the reviewer.       

6.Is the text clear and easy to read?

Yes, but minor English editing is required.  We have further edited our submission. 

Other comments:

On my opinion the article is interesting and well written. However, I highlighted some minor issues:

  • English language: Minor spell check required.  We have spell-checked the revised manuscript.
  • Summary of abbreviations required. The list of abbreviations is provided in the revised submission (i.e., before the Reference section).
  • Introduction: The Authors may improve this section on the theme of therapies for the treatment of OLP. Allow me to suggest a relevant references to include on the use of platelet-rich fibrin on the treatment of OLP: “https://doi.org/10.1007/s00784-020-03702-w”. We are truly grateful to the reviewer for this information; the suggested article is cited (i.e., citation no. 12) and described in the Introduction (lines 57-60). 
  • Materials and methods: This section has been properly prepared.
  • Results: This section is not clearly written. Please improve.  We have tried to improve this section as best as possible.
  • Discussion: This section has been properly prepared.
  • Conclusion: This section has been properly prepared.

Reviewer 2 Report

Formal Structure . Is Ok

The conclusion

A1 in oral mucosal epithelium of human subjects did not differentiate the in- flammatory conditions of OLP and OC from fibroma. Rather, distinct patterns of stain-

ing for these proteins are suggestive of their differential functional roles in the oral mu

cosa. Further, we observed a significant reduction in normalized staining for GILZ, but not Annexin A1, for OLP and OC compared to fibroma. Given the marked and multifac-eted anti-inflammatory effects of GILZ, its reduction in the microenvironment of OC and OLP specimens could curtail growth of Candida albicans in OC while a similar reduction in OLP could exacerbate the inflammation associated with OLP, aspects that require fur- ther investigation.

 the text  is clear and easy to read

 Major Flaws

Insufficient data

The sample size is insufficient ( n= 5 control;  OC 10  and Lichen planus 10 )

Data collection needs to be clearer  Not all are the same oral tissues( tongue, palate, buccal mucosa……..

  The assessment of the samples should be blinded

Author Response

We are truly grateful to the reviewer for her/his very valuable comments and suggestions that we have carefully considered in revision of our manuscript; revisions are highlighted in BLUE for ease of identification.  We are delighted that the reviewer has found our submission to be timely and valuable and are thankful for his/her supportive comments. Items requiring a response appear below.

As an additional comment, we wish to indicate that we have revised Figure 2 to show the 50 μm (i.e., higher magnification) image, only.  [In other words, we have removed the 100 μm, lower magnification, image which appeared in our initial submission.]  We believe that the 50 μm image is sufficient to show candida infection.  We hope this revision is acceptable to the reviewer.      

The sample size is insufficient ( n= 5 control;  OC 10  and Lichen planus 10)  As indicated, we used 10 biopsy samples for each of our experimental conditions (i.e., OC and OLP) and 5 control tissue samples.  Importantly, despite demographic heterogeneity, cellular localization and expression profiles of proteins of interest were similar among samples in each group; thus, it is unlikely that incorporation of additional samples would change study outcomes.  We believe our novel observations lay the foundation for investigation of the role of these proteins in these oral pathologies. 

Data collection needs to be clearer  Not all are the same oral tissues( tongue, palate, buccal mucosa……..  Information regarding site of lesion is provided in Table 1. 

 The assessment of the samples should be blinded.  All samples were coded and assessed prior to data compilation.  

Reviewer 3 Report

Dear Authors,

your paper entitled “Expression Profiles of GILZ and Annexin-A1 in Human Oral Candidiasis and Lichen Planus” investigated the status of GILZ and Annexin-A1 in OLP and candidiasis from paraffin-embedded biopsy samples.

The whole paper is well written, clear, and complete.

M&M section: precise, clear, and detailed enough to reproduce the study. 

Figures: I really appreciated the quality of the pictures and their presentation. I suggest authors add a few explicative lines in the legends to improve the appeal of the usage of pictures for future readers. 

Results
The depictions of the histological findings are a pleasure to read: explicit and figurative. Even though it should be obvious, it is not always possible to read papers so clear and precise. 

Table 1: it was not clear: among the OC enrolled subjects, does “clinical impression/diagnosis” refer to the suspect before histological assessment, that revealed then OC? 
The same goes for OLP and controls.. please, provide a caption to the table to better explain the table and control the format for the table according to the journal’s guidelines. 

The work showed differential expression profiles for GILZ and Annexin A1 in inflammatory oral lesions, suggesting distinct functional roles in human oral mucosa and the opposite significance of their increase/decrease in OC than OLP.
Please, deepen this concept by explaining the clinical implications of these findings (diagnosis, screening..).

Lines 322-324,  332-334: please, remove the bold 

Lines 363: authors reported the altered expression of E-cadherin. They should also refer to the following works (not all mandatory): 

Pannone G, et al.  The role of E-cadherin down-regulation in oral cancer: CDH1 gene expression and epigenetic blockage. Curr Cancer Drug Targets. 2014;14(2):115-27. doi: 10.2174/1568009613666131126115012. PMID: 24274398.

Lo Muzio L, et al.  P-cadherin expression and survival rate in oral squamous cell carcinoma: an immunohistochemical study. BMC Cancer. 2005 Jun 21;5:63. doi: 10.1186/1471-2407-5-63. PMID: 15967043; PMCID: PMC1185522.

Giannelli G, et al. Altered expression of integrins and basement membrane proteins in malignant and pre-malignant lesions of oral mucosa. J Biol Regul Homeost Agents. 2001 Oct-Dec;15(4):375-80. PMID: 11860227.

In conclusion, the work is engaging, well-written, and requires a few minor revisions consisting as follows:
-      Improve the figure legend descriptions
-      Add a legend to table 1 
Improve the discussions on clinical implications, significance, and future perspectives associated with these findings. 

Best regards. 

Author Response

We are truly grateful to the reviewer for her/his very valuable comments and suggestions that we have carefully considered in revision of our manuscript; revisions are highlighted in BLUE for ease of identification.  We are delighted that the reviewer has found our submission to be timely and valuable and are thankful for his/her supportive comments.  Items requiring a response appear below. 

As an additional comment, we wish to indicate that we have revised Figure 2 to show the 50 μm (i.e., higher magnification) image, only.  [In other words, we have removed the 100 μm, lower magnification, image which appeared in our initial submission.]  We believe that the 50 μm image is sufficient to show candida infection.  We hope this revision is acceptable to the reviewer.      

Figures: I really appreciated the quality of the pictures and their presentation. I suggest authors add a few explicative lines in the legends to improve the appeal of the usage of pictures for future readers. We are grateful for your positive and valuable comments; we have provided additional description in the figure legends and the legend for Table 1.

Results
The depictions of the histological findings are a pleasure to read: explicit and figurative. Even though it should be obvious, it is not always possible to read papers so clear and precise. 

Table 1: it was not clear: among the OC enrolled subjects, does “clinical impression/diagnosis” refer to the suspect before histological assessment, that revealed then OC? Yes, clinical impression refers to the clinical evaluation by the examining clinician before obtaining biopsy specimen for histopathological assessment.  This is now further clarified in the legend for Table 1 and the last sentence above Table 1 legend (i.e., line 92).
The same goes for OLP and controls.. please, provide a caption to the table to better explain the table and control the format for the table according to the journal’s guidelines. Thank you!  Your recommendation is incorporated (i.e., Table 1 and its legend). 

The work showed differential expression profiles for GILZ and Annexin A1 in inflammatory oral lesions, suggesting distinct functional roles in human oral mucosa and the opposite significance of their increase/decrease in OC than OLP.
Please, deepen this concept by explaining the clinical implications of these findings (diagnosis, screening..). We now provide further clinical implications (lines 376-382). 

Lines 322-324,  332-334: please, remove the bold  Addressed

Lines 363: authors reported the altered expression of E-cadherin. They should also refer to the following works (not all mandatory): We are grateful to the reviewer for this information.  We have evaluated the suggested publications which are focused on the role of cadherins and integrins in relation to malignant oral lesions without reference to Annexin-A1.  On the other hand, the segment referred to by the reviewer relates to our discussion of the role of Annexin-A1 in oral candidiasis based on the cited literature.  Thus, we are concerned that discussion of the suggested references could be disruptive to the flow of concepts and create confusion for the readership.  We respectfully request reviewer’s due consideration of our concern. 

Pannone G, et al.  The role of E-cadherin down-regulation in oral cancer: CDH1 gene expression and epigenetic blockage. Curr Cancer Drug Targets. 2014;14(2):115-27. doi: 10.2174/1568009613666131126115012. PMID: 24274398.

Lo Muzio L, et al.  P-cadherin expression and survival rate in oral squamous cell carcinoma: an immunohistochemical study. BMC Cancer. 2005 Jun 21;5:63. doi: 10.1186/1471-2407-5-63. PMID: 15967043; PMCID: PMC1185522.

Giannelli G, et al. Altered expression of integrins and basement membrane proteins in malignant and pre-malignant lesions of oral mucosa. J Biol Regul Homeost Agents. 2001 Oct-Dec;15(4):375-80. PMID: 11860227.

In conclusion, the work is engaging, well-written, and requires a few minor revisions consisting as follows:
-      Improve the figure legend descriptions Addressed
-      Add a legend to table 1 Addressed
Improve the discussions on clinical implications, significance, and future perspectives associated with these findings. Addressed

Round 2

Reviewer 2 Report

The paper is well written and is now easier to read for journal readers.
The changes have been made successfully